# Seaweed Extract as a Biostimulant Agent to Enhance the Fruit Growth, Yield, and Quality of Kiwifruit

Vishal Singh Rana [1], Varsha Sharma [1], Sunny Sharma [2,*], Neerja Rana [3], Vijay Kumar [1], Umesh Sharma [4], Khalid F. Almutairi [5], Graciela Dolores Avila-Quezada [6], Elsayed Fathi Abd_Allah [5] and Kasahun Gudeta [7,8,*]

1 Department of Fruit Science, College of Horticulture, Dr. Yashwant Singh Parmar University of Horticulture and Forestry, Nauni 173230, Himachal Pradesh, India
2 Department of Horticulture, School of Agriculture, Lovely Professional University, Phagwara 144411, Punjab, India
3 Department of Basic Sciences, College of Forestry, Dr. Yashwant Singh Parmar University of Horticulture and Forestry, Nauni 173230, Himachal Pradesh, India
4 Department of Tree Improvement and Genetic Resources, College of Forestry, Dr. Yashwant Singh Parmar University of Horticulture and Forestry, Nauni 173230, Himachal Pradesh, India
5 Plant Production Department, College of Food and Agricultural Sciences, King Saud University, P.O. Box 2460, Riyadh 11451, Saudi Arabia
6 Facultad de Ciencias Agrotecnológicas, Universidad Autónoma de Chihuahua, Chihuahua 31350, Mexico
7 School of Biological and Environmental Sciences, Shoolini University of Biotechnology and Management Sciences, Solan 173229, Himachal Pradesh, India
8 School of Applied Biology, Adama Science and Technology University, Adama P.O. Box 1888, Ethiopia
* Correspondence: sunny.29533@lpu.co.in (S.S.); kggutema@shooliniuniversity.com (K.G.)

**Abstract:** The kiwifruit [*Actinidia deliciosa* (A. Chev.) C.F. Liang & A.R. Ferguson] has attained significant importance for commercial cultivation in the mid-Himalayan region of the Indian subcontinent during the last three decades. The fruit quality matching international standards has remained a concern. Keeping in mind the bio-stimulatory effects of seaweed extract, a marine bioactive component in horticultural crops, the current study conducted to elucidate the impact of seaweed extract on kiwifruit growth, yield, and quality was conducted in the Department of Fruit Science's kiwifruit block at Dr. Yashwant Singh Parmar University of Horticulture and Forestry, Nauni, Himachal Pradesh, India. For the studies, nine-year-old Allison kiwi vines of uniform size and vigor were planted at a spacing of 4 m × 6 m. With 11 treatments, the experiment was set up in a randomized block design viz, $T_1$: Spray treatment of 1000 ppm (seaweed extract) SWE at fruit set (FS); $T_2$:Spray treatment of 2000 ppm SWE at FS; $T_3$: Spray treatment of 3000 ppm SWE at FS; $T_4$: Spray treatment of 1000 ppm SWE at FS and 10 days after Fruit set (FS); $T_5$: Spray treatment of 2000 ppm SWE at FS and 10 days after FS (DAFS); $T_6$: Spray treatment of 3000 ppm SWE at FS and 10 days after FS; $T_7$: Fruit dip treatment of 1000 ppm SWE at 10 days after FS; $T_8$: Fruit dip treatment @ 2000 ppm SWE at 10 days after FS; $T_9$: Fruit dip treatment @3000 ppm SWE at 10 days after fruit set; $T_{10}$: Fruit dip treatment @ 5 ppm CPPU at 10 days after fruit set; $T_{11}$: Control. The current study compared several seaweed extract treatments, which were applied at various times and concentrations, to N-(2-chloro-4-pyridyl)-N-phenyl-urea (CPPU-5ppm) and untreated control. Seaweed extract (SWE) dip at 3000 ppm 10 days after the fruit set produced significant growth in fruit length and diameter in growing kiwifruit, which was non-significant with CPPU treatment and superior to control. The shape index, fruit weight, and total fruit yield were also found to be the highest with the same treatment. Fruit quality parameters, namely fruit soluble solids contents (SSC) and total sugars, were recorded at a maximum with the SWE Spray dose of 3000 ppm at FS and 10 DAFS. The SSC: Acid ratio and reducing sugars were recorded as the highest with an application of SWE dip at 3000 ppm 10 DAFS. The application of SWE dip at 2000 ppm 10 DAFS) was found to advance the harvesting maturity by 6 days and also exhibited the lowest physiological loss in weight (% PLW) with the highest ascorbic acid content. After 15 days of storage at ambient room temperature (25 ± 2 °C), the application of SWE dip at 3000 ppm 10 DAFS recorded the highest SSC acid ratio and the lowest

titratable acidity. Thus, the application of seaweed extract dip at 3000 ppm 10 days after the fruit set can be recommended to the farmers as an appropriate alternative to the chemical treatment.

**Keywords:** growth; fruiting; maturity; physiological weight loss; quality production; red algae

## 1. Introduction

The kiwifruit belongs to the genus Actinidia and the family Actinidiaceae and is indigenous to China's Yangtze River valley. It is a vigorously growing, woody perennial fruiting vine with scrambling qualities [1]. The kiwifruit is a distinct and appealing fruit. The fruit is egg-shaped, with a dark brown fuzzy exterior and pale green flesh that is semi-translucent. The fruit has a distinctly sweet flavor and scent. Kiwifruit has a flavor that is similar to a blend of strawberry, rhubarb, and gooseberry. Because of the various nutrients included in it, kiwifruit is a nutritionally packed fruit. Vitamin C, E, fiber, carbohydrates, and minerals including calcium, phosphorus, and potassium are all abundant in it [2]. Allison is a prolific bearer and the most promising kiwifruit cultivar in Himachal Pradesh, out of a range of varieties [3]. However, this cultivar has a proclivity for overbearing, which results in small, low-quality fruits. Poor fruit size and quality are the most significant bottlenecks in the commercialization of kiwifruit in India [4]. As a result, our farmers do not receive a fair price for their produce. Fruit size in kiwifruit is affected by several factors, including variety, pollination, leaf–fruit ratio, crop load, irrigation, nutrition, and canopy management. Plant growth regulators are essential for managing various plant growth and developmental processes [5].

Plant growth regulators, on the other hand, are a very effective method of raising crop production and productivity by promoting fruit growth [6]. They have a direct impact on both the quantitative and qualitative elements of fruit development. CPPU [N-(2-chloro-4-pyridyl)-N-phenylurea] one of the most significant growth regulators with cytokinin-like action is commercially used in kiwifruit [7]. Fruits, which are frequently consumed raw, are more susceptible to chemical contamination owing to residual toxicity than grains and pulses [8]. Organic fruit cultivation gained in popularity as a result of customer demand and farmers [4]. Organic farming is becoming increasingly popular among farmers due to its numerous advantages. Organic farming produces healthy crops with a long-term output and is becoming more popular in the market as a result of its superior quality and lack of chemical toxicity [9]. Organic farming has immediate benefits for farmers and may be performed commercially in the kiwifruit industry [10].

Seaweed extract is useful in sustainable agriculture since it is organic and biodegradable. Seaweed products have become increasingly popular in organic farming [11]. Other than trace minerals, vitamins, amino acids, antibiotics, and micronutrients, its extract includes growth-promoting hormones such as auxins, gibberellins, cytokinins, ethylene, and polyamines. At low concentrations, seaweed extracts can induce a variety of physiological plant responses, including increased plant growth, improved blooming and production, and improved nutritional content and shelf life of fruits. These have become popular as biostimulants for a variety of fruits, vegetables, flowers, and grasses production [12–14]. This beneficial effect of seaweed extract on plant development is similar to the action of phytohormones found in it. Multiple growth regulators have been found in seaweed extracts, including cytokinins, auxins, and gibberellins [15]. The foliar spray of seaweed extract is a common method to increase yield in many commercial crops [16]. The foliar application of mineral nutrients offers a quicker method of supplying nutrients to higher plants than other soil application methods [17].

The effect of seaweed extract application on the growth, productivity, and quality of kiwifruit is currently unknown. Whereas, to our knowledge, this is the first study in India to investigate the impact of seaweed extract on kiwifruit development and quality. Keeping

the above facts in consideration, the proposed study was undertaken to investigate the effect of seaweed extract on fruit growth, yield, and quality of kiwifruit.

## 2. Materials and Methods

### 2.1. Study Area and Plant Materials

The experiment was laid out in the Department of Fruit Science, Dr. YS Parmar University of Horticulture and Forestry Nauni, Solan (HP) during 2018–2020 which is situated at 30.86° N and 77.16° E latitude. The experimental field is located in Himachal Pradesh's sub-temperate, sub-humid, and mid-hills agro-climatic zones (Zone-II). The typical annual rainfall in the study region is between 100 and 130 cm, with most of it falling between July and September. The soil texture in the plowed layer (0 to 20 cm) was determined to be sandy loam. The physico-chemical properties of orchard soil were reported as pH 6.75, electrical conductivity 0.15 dS $m^{-1}$, and organic carbon 0.82% before the experiment began. The available N, P, and K in surface soil were 260.55, 38.00, and 258.35 kg $ha^{-1}$, respectively. For the experiment, kiwifruit vines of the cultivar 'Allison' were carefully selected and planted in 2009, T-bar trained, with rows oriented north-south at a spacing of 4.0 m × 6.0 m (416 vines $ha^{-1}$), the female: male ratio was 9:1.

### 2.2. Experimental Design

During the experiment, the plants were treated with uniform cultural practices. The experiment was laid out in randomized block design (RBD) with three replicas. In each treatment, four vines were selected (44 vines). We marked the four shoots in all directions for analysis purposes. The experiment consisted of 11 treatments: (1) $T_1$: Spray treatment of 1000 ppm (Seaweed Extract) SWE at fruit set (FS); (2) $T_2$: Spray treatment of 2000 ppm SWE at FS; (3) $T_3$: Spray treatment of 3000 ppm SWE at FS; (4) $T_4$: Spray treatment of 1000 ppm SWE at FS and 10 days after FS; (5) $T_5$: Spray treatment of 2000 ppm SWE at FS and 10 days after FS; (6) $T_6$: Spray treatment of 3000 ppm SWE at FS and 10 days after FS; (7) $T_7$: Fruit dip treatment of 1000 ppm SWE at 10 days after FS; (8) $T_8$: Fruit dip treatment @ 2000 ppm SWE at 10 days after FS; (9) $T_9$: Fruit dip treatment @3000 ppm SWE at 10 days after fruit set; (10) $T_{10}$:Fruit dip treatment @ 5 ppm CPPU at 10 days after fruit set; (11) $T_{11}$: Control (untreated). The fruits of selected shoots were dipped and sprayed and 10 random fruits from each vine were selected.

### 2.3. Solution Preparation

As a source of seaweed extract, the company Sea6 Energy Pvt. Ltd. (Banglore, India) employed 'Agrogain,' a patented biostimulant. It is a 100% natural macroalgal extract that works with Tarma-Spurttm, a cutting-edge technology helping plants in producing maximum yield. This is a processed macro-algal extract of 21% *w/w* min, natural acidity regulator, stabilizer, and aqueous diluent of 79% *w/w* prepared from red algae. However, it contains 80% water, and its dry weight basis contains 50% carbohydrates, 1–3% lipids, and 7–38% minerals. Their protein contents are highly variable (10–47%) with high proportions of essential amino acids variable concentrations of 1000 ppm, 2000 ppm, and 3000 ppm seaweed extract solutions were prepared. The 5 ppm CPPU solution was made immediately by dissolving 5 mL (sitofex) in one liter of water. During May, seaweed extract was sprayed on fruit bunches during the fruit set and 10 days after the fruit set. The 1000 ppm, 2000 ppm, 3000 ppm SWE, and 5 ppm CPPU fruit dip treatments were applied. The treatments were performed on clear, calm mornings to properly moisten the fruit. The quick dip method of dipping was used for each fruit at given intervals. The spraying was conducted with the help of a knapsack sprayer and one liter was used for each vine. The spraying of water was also the same (1 L).

### 2.4. Pattern of Fruit Growth

Fruit growth was recorded in terms of fruit length and diameter at 15-day intervals from 15 days after fruit set to the date of harvest. Ten fruits were marked on each vine to

record the data. Fruit length and diameter of all marked fruits were measured periodically with Digital Vernier calipers (Mitutoyo, Japan) and expressed in mm. The values of fruit length and diameter were plotted on the graph to determine the phases of growth [8,16]. The phenological data of the flowering is given in Table 1.

**Table 1.** Phenological data of flowering.

| Treatment Code | Treatment Details | Date of Initiation of Flowering | | Date of Full Bloom | | Duration of Flowering | |
|---|---|---|---|---|---|---|---|
| | | 2018–19 | 2019–20 | 2018–19 | 2019–20 | 2018–19 | 2019–20 |
| $T_1$ | 1000 ppm SWE spray at FS | 25/04 | 01/05 | 28/04 | 04/05 | 10 | 11 |
| $T_2$ | 2000 ppm SWE spray at FS | 24/04 | 28/04 | 27/04 | 01/05 | 09 | 11 |
| $T_3$ | 3000 ppm SWE spray at FS | 26/04 | 02/05 | 30/04 | 06/05 | 11 | 10 |
| $T_4$ | 1000 ppm SWE spray at FS and 10 DAFS | 25/04 | 04/05 | 28/04 | 09/05 | 10 | 10 |
| $T_5$ | 2000 ppm SWE spray at FS and 10 DAFS | 23/04 | 03/05 | 27/04 | 06/05 | 09 | 10 |
| $T_6$ | 3000 ppm SWE spray at FS and 10 DAFS | 26/04 | 30/04 | 30/04 | 05/05 | 09 | 10 |
| $T_7$ | 1000 ppm SWE dip at 10 DAFS | 24/04 | 29/04 | 26/04 | 03/05 | 11 | 08 |
| $T_8$ | 2000 ppm SWE dip at 10 DAFS | 26/04 | 01/05 | 29/04 | 06/05 | 11 | 09 |
| $T_9$ | 3000 ppm SWE dip at 10 DAFS | 26/04 | 04/05 | 28/04 | 06/05 | 11 | 09 |
| $T_{10}$ | 5 ppm CPPU dip at 10 DAFS | 27/04 | 03/05 | 30/04 | 06/05 | 10 | 10 |
| $T_{11}$ | Control (Untreated) | 23/04 | 30/04 | 26/04 | 08/05 | 09 | 10 |

*2.5. Yield and Physical Parameters*

The fruits were harvested during mid-October at the commercial harvest stage and adjudged by attaining a soluble solids content of 6.2%. The yield was expressed in terms of kg/vine and kg/ha. The grade-wise fruit yield was determined by categorizing the harvested fruits into 3 grades based on fruit weight. These grades were A grade (>80 g), B grade (50–80 g), and C grade (<50 g). Ten fruits were randomly picked from each vine under different replications of treatments, and the average fruit length and diameter were measured using a digital vernier caliper. The length and diameter of the fruit were measured in centimeters (cm). The L/D ratio was computed by dividing the length of the fruit by the diameter of the fruit. Similarly, ten fruits from each tree were chosen at random from separate replications of treatments, and the average fruit weight was calculated using physical balancing. The weight of the typical fruit was measured in grams (g).

*2.6. Fruit Chemical Parameters*

The total soluble solids of 10 randomly selected fruits were determined using an Erma-hand refractometer (0–32° Brix). The soluble solids contents were expressed in percent. The titratable acidity was estimated by the standard method as described Association of Official Analytical Collaboration [18]. The sugar to acid ratio was calculated by dividing SSC (%) with titratable acidity (%). Lane and Eynon's volumetric method was employed to estimate reducing sugar content, while the ascorbic acid (Vitamin C) was determined by using the AOAC [18] method. The fruits treated with different treatments were subjected to 15 days of storage at ambient temperature (25 ± 2 °C) and were analyzed for SSC, titratable acidity, and SSC: acid ratio with the above-mentioned methods.

*2.7. Physiological Loss in Weight (PLW)*

On the days of the fruit harvest, the weight of the fruits was measured using physical balance. After 15 days, the fruits were weighed again, and physiological loss was calculated by the formula:

$$\text{Physiological loss in weight (PLW) \%} = \frac{\text{Loss in weight [Initial weight (gm)} - \text{final weight (gm)]}}{\text{Initial fruit weight at harvest}} \times 100$$

*2.8. Economic Analysis*

By comparing the net benefit of various treatments to that of the control, the economic viability of various treatments was determined. The current price of kiwifruit per grade, i.e., "A" grade @ 2.66 US $ per kg, "B" grade @ 2.00 US $ per kg, and "C" grade @ 1.33 US $ per kg, was used for this purpose. The cost of each treatment was estimated by considering the chemical cost, as well as other management expenditures such as labor, manure and fertilizers, irrigation, and pruning, which were all the same and hence were not included in the cost calculation. By deducting the total cost from the net return per vine, the percent increase in net benefit above the control was computed.

*2.9. Statistical Analysis*

The recorded data were run using MS-Excel and SPSS-21 software (SPSS, Chicago, IL, USA) as per the design of the experiment. Analysis of Variance was statistically determined according to the procedure for analysis of Randomized Block Design (RBD).

## 3. Results

*3.1. Impact of Seaweed Extract on the Pattern of Fruit Growth*

A detailed study was carried out to elucidate the effect of seaweed extract on the pattern of fruit growth in terms of fruit length and diameter in kiwifruit starting from 15 days after fruit set until harvest. Figures 1 and 2 show data on the pattern of fruit growth in terms of fruit length and diameter of growing kiwifruit. There are three stages to this double sigmoidal growth curve of kiwifruit development. From fruit set to 75 days following the fruit set, the stage-I of fruit growth was labeled by a fast rise in kiwifruit fruit length and diameter. After that, the fruits went through a period of moderate growth in terms of length, known as stage II, which lasted 45 days. After stage II, the fruit entered stage III, where the length of the fruit increased linearly to 180 days after the fruit set. The increase in fruit diameter was continuous (Figure 2), and uniform, and the highest fruit diameter was measured in fruits treated with CPPU @ 5 ppm 10 DAFS. However, among all the seaweed extract treatments, the application of SWE dip @ 2000 ppm 10 DAFS recorded the maximum fruit diameter.

*3.2. Impact of Seaweed Extract on the Fruit Yield Parameters of Kiwifruit*

The data on the effect of seaweed extract on the total fruit yield of kiwifruit are presented in Table 2. It is evident from the results that different seaweed extract treatments had significant effects on the total fruit yield/vine as compared to the control. The highest fruit yield/vine, total fruit yield/ha, the yield of 'A' grade, and the percent increase in fruit yield over control were recorded with the application of CPPU @ 5 ppm 10 DAFS (Table 2). The fruits treated with SWE spray @ 3000 ppm at FS and 10 DAFS gave the highest yield of 'B' grade fruits. The proportion of 'C' grade fruits was observed to be the highest in untreated fruits. Applications of seaweed extract exhibited significant differences in fruit length and fruit diameter of kiwifruit. Among different seaweed extract treatments, the highest fruit length was recorded with SWE dip @ 3000 ppm 10 DAFS. However, the highest fruit diameter was in SWE dip @ 2000 ppm 10 DAFS which showed non-significant differences with other seaweed extract treatments. Our results revealed that the application of seaweed extract did not exhibit any significant difference in the shape index of kiwifruit. The highest L/D ratio was observed with the application of CPPU dip @ 5 ppm 10 DAFS. The results in the present investigation indicated that the application of seaweed extract had increased the kiwifruit size significantly over control but less from the CPPU and SWE dip @ 3000 ppm 10 DAFS was found as effective as CPPU dip @ 5 ppm 10 DAFFB. It is clear from the data that the average fruit weight was significantly increased with different treatments of seaweed extract over control (Table 2).

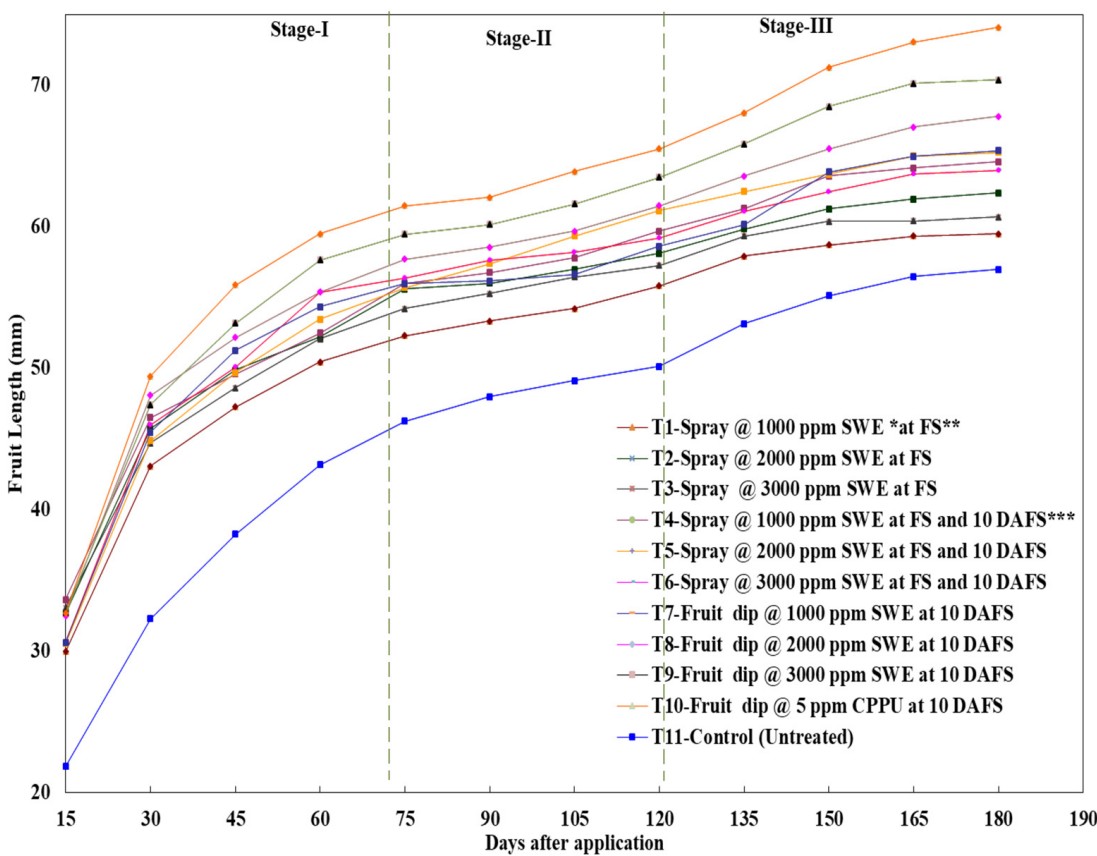

**Figure 1.** Impact of seaweed extract on the fruit growth pattern of kiwifruit in terms of length. SWE *—Seaweed extract, FS **—fruit set, DAFS ***—days after fruit set.

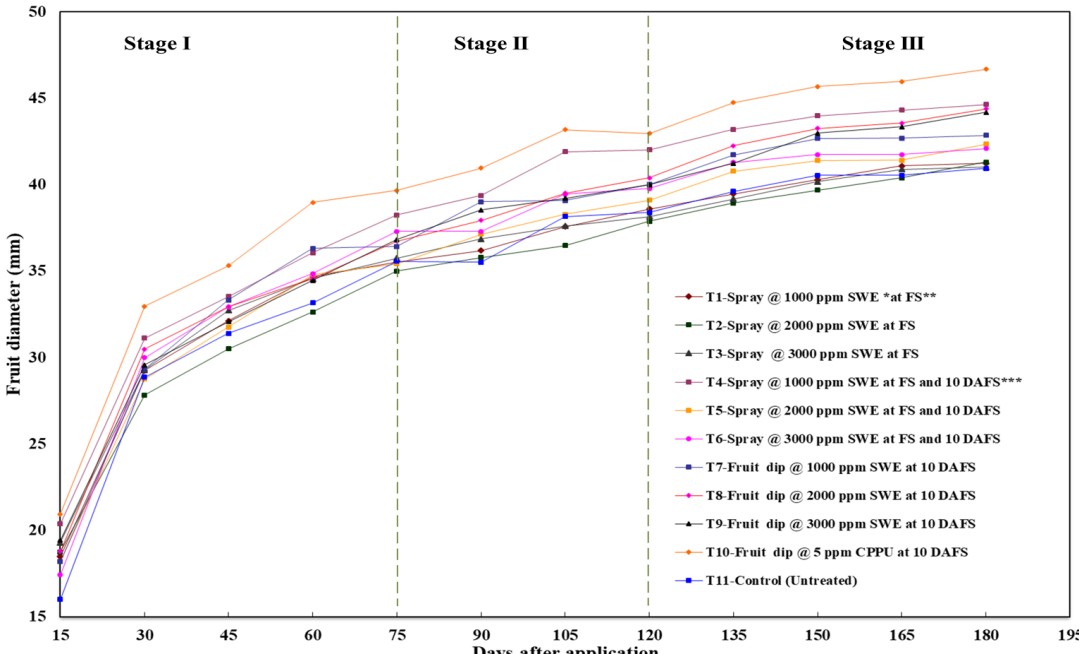

**Figure 2.** Impact of seaweed extract on the fruit growth pattern of kiwifruit in terms of diameter. SWE *—Seaweed extract, FS **—fruit set, DAFS ***—days after fruit set.

**Table 2.** Effect of seaweed extract on total fruit yield in kiwifruit.

| Code | Treatment Details | Total (kg/Vine) | Yield T/ha (%Increase in Yield over Control) | Graded Yield | | | Shape Index | | | Fruit Weight (g/Fruit) |
|---|---|---|---|---|---|---|---|---|---|---|
| | | | | A (kg) | B (kg) | C (kg) | Fruit Length (cm) | Fruit Diameter (cm) | L/D Ratio | |
| $T_1$ | 1000 ppm SWE spray at FS | 25.15 gh | 10.46 g (1.9) | 4.53 h | 8.94 fg | 11.68 b | 5.98 g | 4.10 ef | 1.45 | 52.06 h |
| $T_2$ | 2000 ppm SWE spray at FS | 26.53 fg | 11.03 f (7.49) | 5.13 h | 16.70 bc | 4.70 e | 6.29 ef | 4.13 def | 1.52 | 58.10 f |
| $T_3$ | 3000 ppm SWE spray at FS | 30.10 d | 12.55 d (22.32) | 8.04 f | 17.06 b | 5.09 cd | 6.09 fg | 4.07 f | 1.50 | 55.29 g |
| $T_4$ | 1000 ppm SWE spray at FS and 10 DAFS | 27.39 ef | 11.36 ef (10.65) | 6.97 g | 15.70 d | 4.64 ef | 6.42 de | 4.30 bcd | 1.49 | 61.66 e |
| $T_5$ | 2000 ppm SWE spray at FS and 10 DAFS | 28.14 e | 11.71 e (14.05) | 7.17 g | 15.63 d | 5.35 c | 6.54 cd | 4.26 cde | 1.49 | 63.75 de |
| $T_6$ | 3000 ppm SWE spray at FS and 10 DAFS | 31.45 cd | 13.08 c (27.43) | 9.36 e | 18.75 a | 3.34 g | 6.35 de | 4.26 cde | 1.53 | 64.57 d |
| $T_7$ | 1000 ppm SWE dip at 10 DAFS | 31.40 cd | 13.06 c (27.22) | 10.60 d | 15.90 cd | 4.90 de | 6.55 cd | 4.29 bcd | 1.52 | 62.66 de |
| $T_8$ | 2000 ppm SWE dip at 10 DAFS | 32.50 bc | 13.52 bc (31.68) | 16.67 c | 11.47 e | 4.36 f | 6.70 c | 4.45 ab | 1.50 | 75.50 c |
| $T_9$ | 3000 ppm SWE dip at 10 DAFS | 33.03 b | 13.86 b (35.01) | 19.11 b | 9.55 f | 4.66 ef | 7.05 b | 4.41 bc | 1.59 | 80.66 b |
| $T_{10}$ | 5 ppm CPPU dip at10 DAFS | 35.46 a | 14.75 a (43.67) | 25.91 a | 6.04 h | 3.51 g | 7.41 a | 4.63 a | 1.60 | 84.50 a |
| $T_{11}$ | Control (Untreated) | 24.68 h | 10.26 g | 2.52 i | 8.59 g | 13.57 a | 5.73 h | 4.05 f | 1.40 | 47.77 i |
| | Significance | *** | *** | *** | *** | *** | *** | *** | NS | *** |

Different letters within each column indicate significant differences according to Tukey's HSD test. NS, non-significant; *** significant differences Tukey's HSD test at $p \leq 0.05$. Data are given as mean.

Overall, the highest average fruit weight was with vines treated with CPPU dip @ 5 ppm 10 DAFS as compared to all treated vines.

### 3.3. Effect of Seaweed Extract on Chemical Properties of Kiwifruit

A perusal of the data presented in Table 3 revealed that the seaweed extract treatments had significant effects on the chemical properties of kiwifruit at harvest and the storage of 15 days. The highest SSC was obtained with vine treated with SWE spray @ 3000 ppm at FS and 10 DAFS. On the contrary, the vine treated with SWE dip @ 3000 ppm 10 DAFS recorded the lowest titratable acidity. However, the control exhibited the highest titratable acidity. The SSC acid ratio of fruits as affected by different seaweed extract treatments revealed that the highest SSC acid ratio was observed with fruits subjected to vine treated with SWE dip @ 3000 ppm 10 DAFS. Table 3 shows the results of the study on the influence of seaweed extract on the sugar content of kiwifruit. Fruits subjected to vine treatment with SWE dip @ 3000 ppm 10 DAFS had the greatest total sugars and reducing sugars content. The results showed that the seaweed extract treatment did not affect the kiwifruit non-reducing sugars.

The fruits were tested for SSC and titratable acidity after 15 days. The data revealed that among all the treatments, the application of CPPU dip @ 5 ppm at 10 DAFS recorded the highest SSC content which was significantly superior to all other treatments. The results also showed that the application of SWE dip in 3000 ppm 10 DAFS had the lowest titratable acidity. However, the highest titratable acidity was observed in the control. The data on the SSC acid ratio of fruits as affected by different seaweed extract treatments after 15 days of storage at ambient temperature revealed that the highest SSC acid ratio was obtained in fruits subject to vine treated with SWE dip @ 3000 ppm at 10 DAFS. The present

investigation revealed that the increase in SSC content and decrease in titratable acidity of seaweed extract-treated fruits was less than control and CPPU-treated fruits during storage.

**Table 3.** Effect of seaweed extract on chemical characteristics of kiwifruit.

| Code | Treatment Details | SSC (%) | | TA (%) | | SSC: Acid Ratio | | Sugars Content (%) | | | AA (mg/100 g) |
|---|---|---|---|---|---|---|---|---|---|---|---|
| | | $A_1$ | $A_2$ | $A_1$ | $A_2$ | $A_1$ | $A_2$ | Total (%) | Reducing (%) | Non-Reducing (%) | |
| $T_1$ | 1000 ppm SWE spray at FS | 11.82 cd | 16.10 b | 1.32 b | 0.88 d | 9.04 f | 18.23 c | 9.96 de | 7.86 d | 1.99 | 77.25 d |
| $T_2$ | 2000 ppm SWE spray at FS | 11.72 d | 16.40 b | 1.26 c | 0.81 e | 9.30 ef | 20.28 ab | 10.07 de | 8.12 cd | 1.85 | 78.67 cd |
| $T_3$ | 3000 ppm SWE spray at FS | 12.23 c | 16.20 b | 1.25 c | 0.88 d | 9.87 d | 18.28 c | 10.31 cd | 8.25 c | 1.95 | 79.15 bcd |
| $T_4$ | 1000 ppm SWE spray at FS and 10 DAFS | 11.9 cd | 15.20 c | 1.24 cd | 0.95 c | 9.59 de | 16.01 e | 10.16 cde | 7.95 cd | 2.10 | 80.64 abc |
| $T_5$ | 2000 ppm SWE spray at FS and 10 DAFS | 12.81 b | 15.17 c | 1.21 cde | 0.94 c | 10.58 c | 16.08 e | 10.29 cd | 8.18 cd | 1.99 | 81.31 abc |
| $T_6$ | 3000 ppm SWE spray at FS and 10 DAFS | 13.37 a | 16.19 b | 1.13 fg | 0.79 e | 11.87 a | 20.60 a | 11.21 a | 9.15 a | 2.16 | 81.82 ab |
| $T_7$ | 1000 ppm SWE dip at 10 DAFS | 12.96 ab | 15.96 b | 1.18 def | 0.81 e | 11.02 b | 19.64 b | 10.55 bc | 8.33 c | 2.11 | 80.62 abc |
| $T_8$ | 2000 ppm SWE dip at 10 DAFS | 12.68 b | 16.30 b | 1.16 efg | 0.81 e | 10.97 bc | 20.05 ab | 10.84 ab | 8.75 b | 1.99 | 82.82 a |
| $T_9$ | 3000 ppm SWE dip at 10 DAFS | 13.05 ab | 16.10 b | 1.12 g | 0.78 e | 11.94 a | 20.42 ab | 11.26 a | 9.24 a | 1.93 | 81.18 abc |
| $T_{10}$ | 5 ppm CPPU dip at10 DAFS | 11.99 cd | 17.59 a | 1.27 bc | 1.02 b | 9.47 de | 17.31 d | 10.06 de | 8.04 cd | 1.91 | 80.41 abc |
| $T_{11}$ | Control (Untreated) | 11.74 d | 16.63 b | 1.4 a | 1.13 a | 8.43 g | 14.72 f | 9.77 e | 7.44 e | 2.22 | 78.88 bcd |
| | Significance | *** | *** | *** | *** | *** | *** | *** | *** | NS | *** |

SSC: Soluble solids content, TA: Titratable acidity; AA: Ascorbic acid; $A_1$: at Harvest $A_2$: At 15 days of storage at ambient room Temperature. Different letters within each column indicate significant differences according to Tukey's HSD test. NS, non-significant; *** significant differences parameters indicated by Tukey's HSD test at $p \leq 0.05$. Data are given as mean.

### 3.4. Effect of Seaweed Extract on the Harvest Maturity

Table 4 shows the results on the influence of several treatments of seaweed extract on the days from full bloom to harvest maturity. Untreated fruits had the highest DFFB required to reach harvesting ripeness. The lowest DFFB to reach harvesting maturity was obtained with CPPU-treated fruits, which hastened maturity by 8 days above control. The most effective seaweed extract therapy was determined to be the administration of SWE spray @ 3000 ppm at FS and 10 DAFS, which have provided an earliness kiwifruit maturity by 6 days. The findings of this study demonstrated that the seaweed extract improves harvest maturity by 6 days earlier when compared to the control group. The treatment with CPPU dip @ 5 ppm 10 DAFS, resulted in the greatest advancement in harvesting maturity.

**Table 4.** Effect of seaweed extract on the net benefit of kiwifruit.

| Code | Treatment Details | Harvest Maturity | Physiological Loss in Weight | * Net Benefit (US $.) Percent Increase in Net Benefit over the Control |
|---|---|---|---|---|
| $T_1$ | 1000 ppm SWE spray at FS | 195.00 a | 9.89 bc | 45.37 h (8.26) |
| $T_2$ | 2000 ppm SWE spray at FS | 192.00 ab | 9.53 c | 53.17 g (26.86) |
| $T_3$ | 3000 ppm SWE spray at FS | 190.00 abc | 8.97 d | 61.81 e (47.48) |
| $T_4$ | 1000 ppm SWE spray at FS and 10 DAFS | 194.00 ab | 8.32 e | 55.96 fg (33.51) |

**Table 4.** *Cont.*

| Code | Treatment Details | Harvest Maturity | Physiological Loss in Weight | * Net Benefit (US $.) Percent Increase in Net Benefit over the Control |
|---|---|---|---|---|
| $T_5$ | 2000 ppm SWE spray at FS and 10 DAFS | 193.00 ab | 8.09 ef | 57.19 f (36.46) |
| $T_6$ | 3000 ppm SWE spray at FS and 10 DAFS | 189.00 bc | 7.68 f | 66.49 d (58.62) |
| $T_7$ | 1000 ppm SWE dip at 10 DAFS | 193.00 ab | 6.93 g | 66.44 d (58.53) |
| $T_8$ | 2000 ppm SWE dip at 10 DAFS | 191.00 abc | 6.58 g | 72.99 c (74.14) |
| $T_9$ | 3000 ppm SWE dip at 10 DAFS | 190.00 abc | 6.79 g | 76.01 b (81.34) |
| $T_{10}$ | 5 ppm CPPU dip at10 DAFS | 186.00 c | 12.36 a | 85.24 a (103.37) |
| $T_{11}$ | Control (Untreated) | 195.00 a | 10.32 b | 41.91 i |
| Significance | | *** | *** | *** |

* Net benefit = gross return − chemical cost-labor cost) (Rs.). Different letters within each column indicate significant differences according to Tukey's HSD test. NS, non-significant; *** significant differences parameters indicated by Tukey's HSD test at $p \leq 0.05$. Data are given as mean.

### 3.5. Effect of Seaweed Extract on Physiological Weight Loss (PLW)

The effects of different treatments on the percent physiological loss in weight of fruits after 15 days of storage at room temperature ($25 \pm 2$ °C) (Table 4). With the application of CPPU dip @ 5 ppm 10 DAFS, the highest physiological loss in weight (PLW) was found. Vine treated with SWE @ 2000 ppm 10 DAFS was determined to be the most successful of all the seaweed extract treatments, with the lowest PLW when compared to the control and CPPU-treated fruits.

### 3.6. Effect of Seaweed Extract on Economic Analysis

The kiwifruit treated with CPPU @ 5 ppm 10 DAFS had the highest net benefit according to the economic analysis used to determine the economic variability of different treatments (Table 4). SWE dip @ 3000 ppm 10 DAFS had the maximum net benefit among seaweed extract-treated fruits. The results on the effect of seaweed extract on the percent increase in net benefit over control indicated that among seaweed extract treatments, kiwifruit treated with SWE dip @ 3000 ppm 10 DAFS over control had the highest increase of 81.34% in net benefit over control. However, the application of CPPU @ 5 ppm 10 DAFS exhibited the highest (103.37%) percent of net benefit over control.

### 4. Discussion

Our study provides more robust results than those previously carried out since our research predicted that the seaweed extract exhibited positive responses in the growth stages of the kiwifruit by extending the fruit shelf life. Rana and Rana [3] also reported that kiwifruit exhibited a double sigmoid growth curve, characterized by rapid growth for 75 days (stage I), a period of slow growth for 15 days (stage II), and a period of enhanced growth for 90 days (stage III). An investigation by Hayat [19] on kiwifruit cvs. Allison and Hayward revealed that the kiwifruit exhibited a characteristic double sigmoidal growth curve and observed a significant increase in fruit size till 75 days after full bloom (stage-I), reduced fruit size growth till 120 days after full bloom (stage II), and linear increase in size till 195 days after full bloom (stage-III). In the present investigation, it was revealed that the applications of seaweed extract have increased the fruit length as compared to untreated fruits but the increase in fruit length was lesser than CPPU-treated fruits. Rana et al. [8] reported that the seaweed extract contains cytokinins and gibberellins which are responsible for the regulation of cell division and cell elongation in plants. The enhancement of cytokinin in the CPPU and SWE-treated fruits might have increased the number of cells

by increasing cell division. The seaweed extract has also been reported to increase the fruit length of grapes [16,20].

The present study on the pattern of fruit growth in terms of fruit diameter at a 15-day interval revealed that an increase in fruit diameter did not exhibit a double sigmoidal growth pattern which is supported by the findings of Jao et al. [21] who also reported that the time course changes during fruit development in five cultivars of kiwifruit namely; Abbott, Bruno, Hayward, Monty, and Greensil do not follow double sigmoid growth curve in terms of fruit diameter. The enhancement in fruit diameter by the seaweed extract applications may be attributed to the fact that seaweed extract contains cytokinins which are responsible for the regulation of cell division in plants. The application of seaweed extract might have increased the cytokinin level of fruits, especially when applied at the initial phase of cell division, causing increased cell numbers which ultimately resulted in increased fruit diameter [22]. In this way, a synergetic effect exhibited by the mixture of cytokinin and gibberellins might have imparted its additional contribution towards cell division and cell elongation.

The present investigation envisaged that the seaweed extract improved the total fruit yield over control, graded yield, and percent proportion. It was noted that the fruit dip application of seaweed extract had shown better fruit yield than the foliar application. This may be attributed to the fact that kiwifruit is the hairy and foliar application of seaweed extract may not have resulted in better absorption of the extract as compared to the direct dipping of fruit when compared to CPPU. Similarly, the increment in total fruit yield may be attributed to the fact that the exogenous application of seaweed extract which contains plant growth regulators such as cytokinin, auxins, gibberellins, and mineral nutrients might have resulted in a higher photo-assimilate supply to the fruits [7]. Similar results on the increment in yield by the application of seaweed extract were also reported by Hameedawi and Malikshah [23] in Figures 1 and 2 Pawar and Rana [24] reported that the mixture of gibberellins and cytokinin induced synergistic effects in increasing fruit size. Thus, the improvement in fruit grade might be the result of cell division and cell elongation caused by the exogenous cytokinin and gibberellins present in seaweed extract.

The application of seaweed extract enhanced fruit length and fruit diameter which might be due to the presence of growth regulators which influence cell division and elongation during the early stages of fruit growth. Similar observations on the increase in fruit size due to seaweed extract application were found by Norrie and Keathley [25] in grapes and Khan et al. [16] in grapes. The use of seaweed extract considerably raised average fruit weight when compared to the control group in this research. The presence of plant growth regulators such as cytokinins, auxins, and gibberellins in seaweed extract might explain the rise in average fruit weight. Cytokinin as a growth regulator is linked to nutrient partitioning with the application of CPPU [24]. The increased availability of cytokinin caused by the application of CPPU may have resulted in a larger concentration of photo-assimilates, increasing fruit weight [7]. Photosynthesis and chlorophyll production might be another factor contributing to increased fruit weight by increasing the concentration of photo-assimilates [16].

The current study's findings revealed that seaweed extract treatments had a substantial impact on kiwifruit SSC. Miniawy et al. [26] found that treating strawberries with seaweed extract at a concentration of 2 mL/L enhanced the total soluble solids. According to Khan et al. [16], spraying seaweed extract over grapes at various stages of development boosted the total soluble solids content. Seaweed extract is made up of a variety of micronutrients that have a role in the creation of proteins, amino acids, carbohydrates, and other compounds. When seaweed extract was administered as a foliar treatment, the micronutrients were absorbed by the leaves, possibly leading to an increase in photo-assimilate production. Increased total soluble solids content came from photo-assimilates transported to the sink (fruit) via the phloem. Micronutrients, on the other hand, have a catalytic effect that improves macronutrient absorption in plant tissues, leading to improved fruit quality by boosting SSC content [3]. The use of seaweed extract at various stages of

fruit development reduced the titratable acidity of the juice substantially. Lower acidity in fruits might be owing to increased sugar buildup, improved sugar delivery into fruit tissues, and the conversion of organic acids to sugars [27]. Another possible cause for limiting the titratable acidity was suggested: fast acid consumption of organic acid in respiration. After applying seaweed extract to grapes, Ismail et al. [28] found a reduction in titratable acidity. The SSC acid ratio in kiwifruit was shown to be considerably increased by seaweed extract in this study. The enzymes found in seaweed extract may be responsible for the rise in the SSC acid ratio [16]. Similarly, Ismail et al. [28] found that adding seaweed extract to 'Roomy Red' and 'Thompson seedless' grapes increased the SSC acidity ratios. Seaweed extract considerably enhanced the SSC acid ratio in strawberries, according to Miniawy et al. [26].

The results showed that the seaweed extract treatment did not affect the kiwifruit non-reducing sugars. The greatest levels of non-reducing carbohydrates were seen in the control group. The application of seaweed extract to kiwifruit had a significant effect on total sugars and lowering sugars, according to this study. According to Hameedawi and Malikshah [23], applying seaweed extract to plants raised the quantity of chlorophyll in the leaves, perhaps increasing the rate of photosynthesis. The rise in total sugars and reducing sugars might be attributed to a faster rate of photosynthesis, which could have resulted in more carbohydrate buildup in fruits. In datepalm (*Phoenix dactylifera* L.) cv. Sukary. Omar, et al. [29] found that seaweed extract produced higher reducing sugars and total sugars than potassium nitrate. Similarly, Roshdy [30] found that using 0.05% seaweed extract boosted overall sugar levels in the Grandnaine banana cultivar. Table 3 shows the ascorbic acid concentration of kiwifruit as a function of the amount of seaweed extract applied. The vine treated with SWE dip @ 2000 ppm 10 DAFS had the maximum ascorbic acid concentration, according to the results. The concentration of ascorbic acid in kiwifruit was shown to be dramatically raised by seaweed extract in this study. Seaweed extract includes betaine, which slows the breakdown of chlorophyll and increases the chlorophyll content of treated plants. The increase in chlorophyll concentration might have resulted in a faster rate of photosynthesis and, as a result, more photo-assimilates being produced. The increase in ascorbic acid concentration might be due to a greater supply of photo-assimilates, which provides more substrate for ascorbic acid production. The findings of Abd El-Motty et al. [31] in the 'Keitte' mango, observed a significant increase in ascorbic acid content over control with a 2% spray of seaweed extract. Enzymes found in seaweed extract boosted the rate of ascorbic acid generation in grapes, according to Khan et al. [16].

The present investigation revealed that the increase in SSC content and decrease in titratable acidity of seaweed extract-treated fruits was less than control and CPPU-treated fruits during storage. The maintenance of SSC might be due to the reduction in the degradation of total sugars present in fruits during storage. Soppelsa et al. [27] reported the positive effect of seaweed extract on the maintenance of total soluble solids in apples during storage. The decrease in titratable acidity might be due to the presence of some micronutrients in seaweed extract which might have resulted in the respiration of organic acid stored in cells and the formation of salts of malic acid. Melo et al. [32] reported similar results in Tommy Atkins cv. of mango. According to Patterson et al. [33], CPPU-treated fruits attained commercial harvest maturity one week before untreated fruits. Because seaweed extract improved the pace of fruit growth, starch accumulated in the fruits at an earlier stage of development. Because of the early buildup of starch in the fruits, treated kiwifruit reached an SSC content of 6.5% earlier than untreated kiwifruit. Seaweed extract treatment enhanced kiwifruit maturity by one week [3]. Fornes et al. [34] discovered that increasing the concentration of seaweed extract accelerates the maturity of Satsuma and Clementina mandarins by 5–7 days.

The current study's findings clearly showed that seaweed extract extended the shelf life of kiwifruit cv. Allison. Fruits treated with seaweed extract had a longer shelf life than fruits treated with CPPU or untreated fruits. Colla et al. [12] found that in addition seaweed

extract to fruits increased the amount of calcium in the fruit. Calcium is recognized to have an important function in cell wall stability. Seaweed extract also contains micronutrients such as zinc and silicon, which, when combined with calcium, strengthen the cell wall structure, possibly resulting in less fruit mass loss and the preservation of kiwifruit physicochemical parameters [34]. Norrie and Keathley [25] discovered that the fruits treated with seaweed extract in grape cv. Thompson tasted better. Seedless performed better in storage than untreated ones. Seaweed extract also decreased PLW and fruit degradation in oranges, according to Omar et al. [24]. According to Melo et al. [32], seaweed extract can be utilized as a substitute for mango preservation.

## 5. Conclusions

The focus of this research was to determine the efficacy of seaweed extract treatments as an organic alternative to CPPU treatments for improving kiwifruit fruit development, production, and quality. The findings of this study demonstrated that the application of seaweed extract altered the double sigmoidal growth curve of kiwifruit during fruit development. The treatment with SWE at the rate of 3000 ppm 10 DAFS was determined to be the best. For all biochemical parameters such as SSC, titratable acidity, sugars, and ascorbic acid, the treatment SWE dip at the rate of 3000 ppm 10 DAFS was determined to be superior to CPPU-treated fruits. When compared to the control, the seaweed extract-treated fruits were harvested 3–6 days earlier. CPPU-treated fruits, on the other hand, were 8 days ahead of control in terms of harvesting ripeness. Furthermore, the fruits treated with seaweed extract behaved better during storage, with the lowest percentage of physiological weight loss and a superior SSC acid ratio. The post-harvest life of CPPU-treated fruits was shown to be lower than that of seaweed extract-treated fruits, resulting in shorter shelf life for kiwifruit. However, CPPU treatment had the maximum net benefit over control (103.04%), followed by SWE drop at the rate of 3000 ppm 10 DAFS, which had an 81.35% net benefit over control. As a result, growers might be advised to apply a seaweed extract dip at the rate of 3000 ppm 10 days after fruit set as a viable alternative to chemical treatment.

**Author Contributions:** Conceptualization, V.S.R., V.S. and S.S.; Data curation, V.S.R. and V.S.; Formal analysis, V.S.R., S.S. and V.K.; Funding acquisition, V.S.R. and S.S.; Investigation, V.S.R., N.R. and V.K.; Methodology, V.S.R. and N.R.; Project administration, V.S., S.S., N.R. and V.K.; Resources, V.S.R., V.S., S.S.; K.G., K.F.A., G.D.A.-Q. and E.F.A.; Software, V.S.R. and V.K.; Supervision, V.S.R., S.S. and V.K.; Validation, V.S.R. and V.S.; Visualization, S.S.; Writing—original draft, V.S.R., V.S., U.S. and S.S.; Writing—review & editing, S.S., U.S., K.F.A., G.D.A.-Q. and E.F.A. All authors have read and agreed to the published version of the manuscript.

**Funding:** The author also would like to extend their sincere appreciation to the Researchers Supporting Project Number (RSPD2023R561), King Saud University, Riyadh, Saudi Arabia.

**Data Availability Statement:** Available data is provided in the publication and any other information will be provided on demand.

**Acknowledgments:** The authors are grateful to the Department of Fruit Science, Dr. YSP University of Horticulture and Forestry Nauni, Solan (HP) for offering research infrastructure for laying out the research trial. The author also would like to extend their sincere appreciation to the Researchers Supporting Project (Project number: RSPD2023R561), King Saud University, Riyadh, Saudi Arabia.

**Conflicts of Interest:** The authors declare no conflict of interest.

## Abbreviations

SWE: Seaweed extract; CPPU: N-(2-chloro-4-pyridyl)-N-phenyl-urea; DAFS: days after fruit set; FS: Fruit set SSC: Soluble solid contents; PPM: parts per millions PLW: Physiological loss in weight; DFFB: Days from full bloom to harvest.

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
