# Peer review of "Seaweed Extract as a Biostimulant Agent to Enhance the Fruit Growth, Yield, and Quality of Kiwifruit"

_horticulturae, doi:10.3390/horticulturae9040432_

Round 1

Reviewer 1 Report (Previous Reviewer 2)

Dear authors/editor,

There are still some points in the manuscript to be improved that the authors did not consider from the previous review.  All the suggestions are given in the PDF.

Author Response

Dear Sir/Madam 

We have addressed your comments and highlighted in the manuscript body . 

Reviewer 2 Report (New Reviewer)

Dear Author

The work is good. Need some minor correction.  As a field experiment, I am confusing about your significance level. Please check the data.

With best regards

Author Response

Dear Sir/Madam 

We have addressed your comments and highlighted in the manuscript body . 

Reviewer 3 Report (Previous Reviewer 3)

This article is better than the first submitted

Author Response

Dear Sir/Madam 

We have addressed your comments and highlighted in the manuscript body. 

Reviewer 4 Report (New Reviewer)

Abstract section:

please avoid such detailed descriptions of the material and methods used: T1: Spray treatment of 1000 ppm (Seaweed Extract) SWE at fruit set 34 (FS); (2)T2:Spray treatment of 2000 ppm SWE at FS; (3) T3: Spray treatment of 3000 ppm SWE at FS; 35 (4) T4: Spray treatment of 1000 ppm SWE at FS and 10 days after Fruit set (FS); (5)T5:Spray treat- 36 ment of 2000 ppm SWE at FS and 10 days after FS (DAFS); (6) T6: Spray treatment of 3000 ppm 37 SWE at FS and 10 days after FS; (7)T7: Fruit dip treatment of 1000 ppm SWE at 10 days after FS; 38 (8)T8:Fruit dip treatment @ 2000 ppm SWE at 10 days after FS; (9) T9: Fruit dip treatment @3000 39 ppm SWE at 10 days after fruit set; (10)T10:Fruit dip treatment @ 5 ppm CPPU at 10 days after fruit 40 set; (11)T11:Control. Please rephrase this part. 

In general, the abstract section is too long.

line 107:add punctuation after yield and quality of kiwifruit.

line 125: correct was laid out to 'established'.

145 and line 153: punctuation. 

line 164: correct kg/ha−1 to kg/ha or kg ha−1

line 166: why not the same 10 fruits as in line 156?

Line 178: delete the bracket at the beginning. 

Lines 208-211 are repeating.

Since there is a separate Discussion section please rename Results and discussion to solely Results. 

Line 360: this sentence is not finished. 

Conclusion section is excelent. 

Author Response

Dear Sir/Madam 

We have addressed your comments and highlighted in the manuscript body.

This manuscript is a resubmission of an earlier submission. The following is a list of the peer review reports and author responses from that submission.

Round 1

Reviewer 1 Report

I think this is a very interesting and extensive study exploring the biostimulant effects of seaweed extract on kiwi fruits and I would like to commend the authors for putting together a wonderful study. Its also interesting to note that findings from the current study agrees with some of the commonly known attributes of biostimulants that includes Line 202-203 where the highest fruit diameter was observed from the lowest biostimulant concentration treatment. Also Line 204-205 shows that the application method can have significant effects on the efficacy of a biostimulant in this case spray over dip.

It would be interesting to know how a combination of both spray and dip treatments at differing biostimulant concentration affect the kiwi fruit quality parameters. 

Reviewer 2 Report

Dear Authors/Editor,

Here I submit the review including text comments in PDF attached. 

My major remarks are regarding improving the experimental design description and the chemicals application methods.

The second is regarding the repetitions in the Discussion section on the effects of bioregulators.

And the third is regarding the duration of the experiment. It is usual that field research (such as this one) affected by environmental factors is performed for at least two years in order to derive reliable conclusions. 

Other remarks are given in the text.

Sincerely, reviewer.

Reviewer 3 Report

 Keywords: Kiwifruit; Seaweed extract; CPPU; Biostimulants; quality production; use words that are not part of the title.

Need improvement on materials and methods; because from what was written this research cannot be repeated because the method is not clear.

T1:Spray treatment of 1000 ppm (Seaweed Extract) SWE at 117 fruit set (FS); (2)T2:Spray treatment of 2000 ppm SWE at FS; (3) T3: Spray treatment of 118 3000 ppm SWE at FS; (4) T4: Spray treatment of 1000 ppm SWE at FS and 10 days after 119 FS; (5)T5:Spray treatment of 2000 ppm SWE at FS and 10 days after FS; (6) T6: Spray 120 treatment of 3000 ppm SWE at FS and 10 days after FS; (7)T7: Fruit dip treatment of 1000 121 ppm SWE at 10 days after FS; (8)T8:Fruit dip treatment @ 2000 ppm SWE at 10 days after 122 FS; (9) T9: Fruit dip treatment @3000 ppm SWE at 10 days after fruit set; Need more information  of the treatments especially the application volume per fruit or number of fruits.

(1)  How many ml or cc of SWE applied per fruit or number of fruits?

(2)  How many fruits per treatment?

(3)  How many vines per treatment?

(4)  How many second the "dip" treatments?

(5)  What  seaweed species is used in this research?

(6)  What is the biological active compound of the seaweed that consider as biostimulant? Is any cytokinin-like compound? Need result of quantitative analysis of chemical properties of seaweed extract.

(line 278-279): The findings of this study demonstrated that the seaweed extract increased  maturity by 6 days when compared to the control group. What “increased maturity” mean?
